# Worsening Renal Function and Adverse Outcomes in Patients with HFpEF with or without Atrial Fibrillation

**DOI:** 10.3390/biomedicines11092484

**Published:** 2023-09-07

**Authors:** Linjuan Guo, Xiaojuan Wu

**Affiliations:** 1Department of Cardiology, Jiangxi Provincial People’s Hospital, The First Affiliated Hospital of Nanchang Medical College, Nanchang 330000, China; 2Department of Gastroenterology, Ganzhou People’s Hospital, Ganzhou 341000, China

**Keywords:** heart failure with preserved ejection fraction, worsening renal function, atrial fibrillation, interaction

## Abstract

Since worsening renal function (WRF) and atrial fibrillation (AF) often coexist in preserved ejection fraction (HFpEF), we aimed to investigate the effect of WRF on the prognosis of HFpEF patients with and without AF. The study population of this study (n = 1763) was based on the subset of the Americas in the Treatment of Preserved Cardiac Function Heart Failure with an Aldosterone Antagonist Trial (TOPCAT). We found that the cumulative probabilities of the primary composite outcome and cardiovascular death were significantly higher in AF patients post-WRF when compared to non-AF patients. In the time-dependent Cox proportional hazard model, WRF was significantly associated with higher risks of adverse outcomes (primary composite outcome: HR = 1.58 (95% CI, 1.19–2.11); all-cause death: HR = 1.50 (95% CI, 1.10–2.06); cardiovascular death: HR, 2.00 (95% CI, 1.34–3.00)) after adjustments for confounding factors at baseline in HFpEF patients with AF, whereas in HFpEF patients without AF, WRF was not significantly associated with any adverse outcome. *p* for interactions for the primary composite outcome, cardiovascular death, and AF were significant. In conclusion, these findings highlight that WRF was associated with a greater risk of the primary composite outcome, all-cause death, and cardiovascular death in HFpEF patients with AF.

## 1. Introduction

Heart failure (HF) is associated with high morbidity and mortality [1]. As a heterogeneous syndrome, HF patients usually suffer from diverse comorbidities, among which worsening renal function (WRF) has a relatively high incidence (approximately 50% in HF patients) [2,3]. According to the Study of Left Ventricular Dysfunction (SOLVD) trial, WRF is associated with higher risks of all-cause mortality, the combined endpoint of death or hospitalization for HF, and pump-failure death [4]. In the Second Prospective Randomized Study of Ibopamine on Mortality and Efficacy (PRSIME II) trial, HF patients with reduced estimated glomerular filtration rate (eGFR) (44 mL/min, consistent with underlying WRF) showed a three-fold greater risk of mortality than HF patients without reduced eGFR [5]. In addition, several studies have shown that adverse effects of WRF on the prognosis of HF patients were irrespective of left ventricular ejection fraction (LVEF) < 35% [3,6].

Heart failure with preserved ejection fraction (HFpEF), characterized by the LVEF of HF patients > 50%, accounts for 50% of the total HF cases [7,8]. Atrial fibrillation (AF) is one of the most common cardiac arrhythmias that affects millions of patients worldwide [9]. According to the EuroHeart Failure survey, 34% of HF patients suffered from AF before being hospitalized, whereas 9% of the patients were diagnosed with AF during hospitalization [10]. As a common comorbidity of HFpEF, AF has a strong correlation with age and shares common clinical features with HFpEF [11]. Additionally, AF is an independent predictor for adverse prognosis in HFpEF patients [12]. Both WRF and AF are common comorbidities in the development of HFpEF, and a number of HFpEF patients suffer from both WRF and AF. However, the overall impact of the two comorbidities on HFpEF patients remains unclear. Therefore, our current study aimed to explore the effect of WRF on the prognosis of HFpEF patients with and without AF.

## 2. Materials and Methods

### 2.1. Study Population

We included the study population in the Treatment of Preserved Cardiac Function Heart Failure with an Aldosterone Antagonist Trial (TOPCAT; https://clinicaltrials.gov. accessed on 12 January 2023. Unique identifier: NCT00094302. Unique identifier: NCT00094302), a randomized, double-blind, multicenter, phase III clinical trial that involved 3445 HFpEF patients with a mean follow-up of 3.3 years [13]. The objective of this trial was to assess the effect of spironolactone in improving the prognosis of HFpEF patients when compared to a placebo [13]. In brief, all the included participants were ≥50 years and had at least one symptom and a sign of HF. They were from Georgia, Russia, or the Americas (Brazil, Argentina, Canada, United States), characterized by an LVEF ≥ 45%, a serum potassium level of < 5 mmol/L, a systolic blood pressure < 140 mmHg, and either elevated brain natriuretic peptide (BNP) within two months or a previous history of HF hospitalization within a year. Patients would be excluded if they took other aldosterone antagonists within 14 days before the trial, had a life expectancy of <3 years, or suffered from severe renal dysfunction (serum creatinine ≥2.4 mg/dL or eGFR < 30 mL/min/1.73 m^2^).

Baseline characteristics of patients including demographics, morbidities of comorbid conditions, and medication treatment significantly varied due to geographic disparities [14,15,16]. To keep data consistent and reduce errors caused by different baseline characteristics, the population of this study was restricted to the Americas. We excluded patients due to the lack of a record of AF history. Finally, 1763 patients with HFpEF were involved in the present analysis. The findings of this study were reported according to the STROBE (Strengthening the Reporting of Observational Studies in Epidemiology) statement.

The TOPCAT trial was approved by the institutional review board locally. All the participants signed informed consent. The dataset used in this study was acquired from the National Heart, Lung, and Blood Institute (NHLBI). Our current study was approved by Jiangxi Provincial People’s Hospital, the First Affiliated Hospital of Nanchang Medical College.

### 2.2. Definition of WRF

WRF was defined as a ≥30% decline in eGFR during follow-up. Compared with kidney failure and doubling of creatinine levels, which were late events in chronic kidney diseases, a ≥30% decline in eGFR occurred more commonly and at earlier stages of kidney diseases. The National Kidney Foundation and the Food and Drug Administration co-sponsored a scientific workshop to determine whether alternative GFR-based endpoints could be used in clinical trials and concluded that a 30% to 40% decline in eGFR over two to three years might be an acceptable surrogate endpoint [17].

### 2.3. Study Outcomes

Consistent with the TOPCAT trial, the primary outcome was a composite of HF hospitalization, aborted cardiac arrest, and cardiovascular death. The secondary outcomes were all-cause death, cardiovascular death, HF hospitalization, stroke, myocardial infarction, and AF (referred to as new-onset AF in HFpEF patients without a history of AF or AF recurrence in HFpEF patients with AF). The definitions of the studied outcomes were applied from the original TOPCAT trial. During the follow-up, the outcomes of this study were recorded through subject contacts or by interview and medical record review at the clinic site. The Clinical Endpoints Center independently adjudicated the event of each outcome.

### 2.4. Follow-Up Method

As reported previously, in the TOPCAT trial, the follow-up visits to monitor symptoms, medications, and events and to dispense the study drug were planned every four months during the subject’s first year of the study, and every six months thereafter. Data on participants who did not have an event of time-to-event outcomes were censored at the date of last available follow-up information for clinical events.

### 2.5. Statistical Analysis

Baseline patient characteristics were displayed according to the history of AF. Continuous variables were presented in the form of mean with standard deviation (normal distribution) or medians with interquartile ranges (nonmoral distributions). Categorical variables were expressed as numbers and percentages. For continuous variables, differences between groups were investigated using unpaired Student’s *t*-tests (normal distribution) or the Wilcoxon–Mann–Whitney test (non-normal distribution). As for categorical variables, we conducted chi-square tests or the Kruskal–Wallis test. Incidence rates with 95% confidence intervals (CIs) and Kaplan–Meier survival analyses were performed to evaluate the crude cumulative probability of primary and secondary outcomes prior to or post-WRF. Both crude and multivariable-adjusted time-dependent Cox proportional hazard models were performed to assess the impact of WRF and AF on adverse outcomes, displayed as hazard ratios (HRs) and 95% CIs. The significantly different variables (age, sex, race, treatment group, smoking status, New York Heart Association (NYHA) functional class, BMI, SBP, diabetes mellitus, aspirin, warfarin, long-acting nitrate treatment, diuretics, history of PCI, and pacemaker implantation) at baseline were adjusted. *p* for interaction was calculated to test the possible relationship between the effect of WRF and AF on adverse outcomes.

All the analyses were performed using R (version 4.1.0; R Foundation for Statistical Computing). Two-tailed *p*-values of <0.05 were considered statistically significant.

## 3. Results

### 3.1. Baseline Patient Characteristics

Among all the participants of this study, 743 patients (42.1%) had a history of AF (Table 1) with a mean age of 71.5 ± 9.7 years. The mean age of AF patients was significantly higher than non-AF patients (74.0 ± 8.6 versus 69.6 ± 10.0 years). AF patients had higher proportions of males and white race but a relatively lower body mass index. The eGFR was markedly lower in AF patients, and a higher proportion of non-AF patients were current smokers and suffered from WRF. The prevalence of diabetes mellitus was significantly greater in the non-AF group. The two studied groups had similar proportions of the NYHA functional class levels. Compared with those without AF, more patients with AF used pacemakers and took diuretic, aspirin, long-acting nitrate, or warfarin.

### 3.2. Cumulative Probability of Adverse Outcomes

In the Kaplan–Meier survival analyses, the cumulative probabilities of all the adverse outcomes prior to WRF were comparable between HFpEF patients with and without AF (Figure 1). In the post-WRF group, the cumulative probabilities of the primary composite outcome (*p*-value = 0.0083) and cardiovascular death (*p*-value = 0.0096) were significantly higher in AF patients when compared to non-AF patients (Figure 2). Conversely, the cumulative probability of AF post-WRF was higher in HFpEF patients without AF compared with those with AF (*p*-value = 0.009). For other adverse events, the two groups of HFpEF patients presented similar risks of stroke, myocardial infarction, HF hospitalization, and all-cause death (*p*-value > 0.05).

### 3.3. Incidence Rates of Adverse Outcomes

As shown in Table 2, when compared with non-AF patients post-WRF, AF patients showed increased incidence rates of the primary composite outcome (16.5 (95% CI, 11.6–21.5) versus 9.3 (95% CI, 6.0–12.6) per 100 patient-years), all-cause death (95% CI, 11.5 (7.3–15.8) versus 8.0 (95% CI, 4.9–11.0) per 100 patient-years), cardiovascular death (8.3 (95% CI, 4.6–12.0) versus 4.5 (95% CI, 2.2–6.9) per 100 patient-years), stroke (2.6 (95% CI, 0.5–4.7) versus 1.5 (95% CI, 0.1–2.9) per 100 patient-years), myocardial infarction (2.6 (95% CI, 0.4–4.7) versus 1.2 (95% CI, 0–3.1) per 100 patient-years), and HF hospitalization (10.9 (95% CI, 6.7–15.0) versus 7.4 (95% CI, 4.5–10.4) per 100 patient-years). However, compared with non-AF patients post-WRF, patients with AF had a lower incidence rate of AF (0.9 (95% CI, 0.3–2.2) versus 3.7 (95% CI, 1.6–5.9) per 100 patient-years).

### 3.4. Associations of WRF with Adverse Outcomes

As shown in Table 3, in the unadjusted or adjusted analysis, WRF was significantly associated with increased risks of primary composite outcome, HF hospitalization, cardiovascular death, and all-cause death among the overall HFpEF patients. In HFpEF patients with AF, the development of WRF was associated with higher risks of the primary composite outcome (1.64 (95% CI, 1.24–2.16)), all-cause death (1.36 (95% CI, 1.01–1.84)), and cardiovascular death (1.84 (95% CI, 1.26–2.71)) in the unadjusted analysis. These associations remained consistent after adjusting for significantly different variables at baseline (primary composite outcome: HR = 1.58 (95% CI, 1.19–2.11); all-cause death: HR = 1.50 (95% CI, 1.10–2.06); cardiovascular death: HR, 2.00 (95% CI, 1.34–3.00)). In HFpEF patients without AF, WRF was not significantly associated with any adverse outcome including the primary composite outcome (1.10 (95% CI, 0.83–1.46)), HF hospitalization (1.28 (95% CI, 0.93–1.76)), cardiovascular death (1.00 (95% CI, 0.65–1.53)), all-cause death (1.24 (95% CI, 0.91–1.70)), stroke (1.49 (95% CI, 0.70–3.16)), myocardial infarction (1.15 (95% CI, 0.64–2.06)), and AF (1.21 (95% CI, 0.76–1.94)) after adjustments.

In the *p* for interaction analysis, the results of all-cause death, HF hospitalization, stroke, and myocardial infarction were not different between non-AF patients with and without WRF (*p* for interaction > 0.05, Table 3). By contrast, *p* for interactions for the primary composite outcome (*p* for interaction = 0.008), cardiovascular death (*p* for interaction = 0.005), and AF (*p* for interaction = 0.015) were significant.

## 4. Discussion

This study aimed to explore the impact of WRF on the prognosis of HFpEF patients with and without AF. Our results based on the TOPCAT trial suggest that WRF was significantly associated with higher risks of the primary composite outcome, all-cause death, and cardiovascular death in HFpEF patients with AF, whereas WRF was not significantly associated with any adverse outcome in HFpEF patients without AF. *p* for interactions for the primary composite outcome (a composite of HF hospitalization, aborted cardiac arrest, and cardiovascular death), cardiovascular death, and AF were significant. WRF could be a significant risk factor for adverse cardiovascular events in HFpEF patients with AF compared with those without AF.

The heart and kidney interact with each other to maintain the stability of the hemodynamics, blood volume, and vascular tone of the body [18,19]. Any damage that happens in this system could be deleterious, resulting in a cycle of organ dysfunction [19]. In addition, mineralocorticoid receptor antagonists (MRAs) including spironolactone are widely used for treatment in HFpEF patients [15,20]. Although spironolactone effectively improves the prognosis of HFpEF patients, it is associated with an increased WRF risk, potentially resulting in underlying renal injury [16].

The presence of WRF is common in HFpEF patients, which is often correlated with a poor prognosis [3,6,21]. Nowadays, the exact mechanisms regarding how WRF aggravates the progression of HFpEF are still uncertain, which may be multifactorial [20]. It is uncertain whether the activation of these mechanisms directly leads to adverse outcomes in HF caused by WRF or is just a marker of severe HF status. In addition, the impact of WRF on HF may differ by HF phenotype. WRF could be considered as a reflection of disrupted hemodynamics during the interaction between the cardiovascular and renal systems [22]. Moreover, WRF may be a marker of general vascular disease including atherosclerosis in the heart.

AF is associated with atrial fibrosis, impaired atrial function, and the dilation of the left atrium, which could be considered as an independent predictor for the prognosis of HFpEF [12,23]. AF and renal impairment have a bidirectional relationship and the prevalence of chronic renal disease appears higher in patients with AF, whereas acute abnormalities in cardiac function including AF are related to an increased risk of kidney injury [19,24,25,26]. Hence, the impact of WRF and AF on the prognosis of HFpEF patients, and the interaction between the two comorbidities, merit further evaluation.

This study investigated the impact of WRF on the prognosis of HFpEF patients with and without AF as well as the interaction between WRF and AF based on the TOPCAT study. We found that WRF was no longer an independent risk factor in the presence of AF, and could significantly interact with AF in the prognosis of HFpEF. Consistent with the underlying mechanisms in WRF, AF might lead to changes in hemodynamics in HF patients. However, this mechanism generally pertains to HFrEF, the contribution of which to HFpEF needs further investigation [12,27].

As a common comorbidity of HFpEF, AF has a strong correlation with age and shares common clinical features with HFpEF. In addition, the presence of WRF is common in HFpEF patients. In our current study, WRF was correlated with a poor prognosis in HFpEF patients, especially for those with AF. Therefore, our results based on the TOPCAT trial indicated that HFpEF patients with AF need to pay attention to the management of renal function in everyday clinical practice. Of note, the cumulative probability of AF prior to WRF was comparable between HFpEF patients with and without AF, whereas the cumulative probability of AF post-WRF was higher in HFpEF patients without AF compared with those with AF. However, these findings of AF incidence did not remain after adjustments, which needs further investigation.

### Limitations

There were still several limitations in this study. First, the study population was limited to the Americas, limiting the generality of the results due to the underlying regional variation. Second, the distribution of age and race were not uniform within the two studied groups. Although we have adjusted age and race as the confounding factors in the Cox proportional hazard model, potential bias may still occur. Third, based on the TOPCAT study, participants were evenly divided into the spironolactone or placebo groups, and we failed to subdivide the two treatment groups due to the limited data. Spironolactone has been proven to be associated with increased WRF, thus potentially confounding the results. Forth, as the TOPCAT trial did not reproduce the ultrasound studies, e/é and other parameters of diastolic function could not be further analyzed. In addition, duo to lack of relevant data on the BNP during and after the worsening of renal function, we were unable to conduct the relevant analysis. Finally, the TOPCAT trial included a few patients with non-vitamin K antagonist oral anticoagulants, and further study could examine the impact of non-vitamin K antagonist oral anticoagulants on our results [28]. In addition, although some studies suggested hyperuricemia as a marker of HFrEF [29], whether similar results were observed in the HFpEF population needs further examination.

## 5. Conclusions

Based on the TOPCAT trial, our present study demonstrated that WRF was associated with greater risks of the primary composite outcome, all-cause death, and cardiovascular death among HFpEF patients with AF. WRF did not act alone under the presence of AF in HFpEF patients with marked interactions between WRF and AF in the prognosis of HFpEF patients.

## Figures and Tables

**Figure 1 biomedicines-11-02484-f001:**
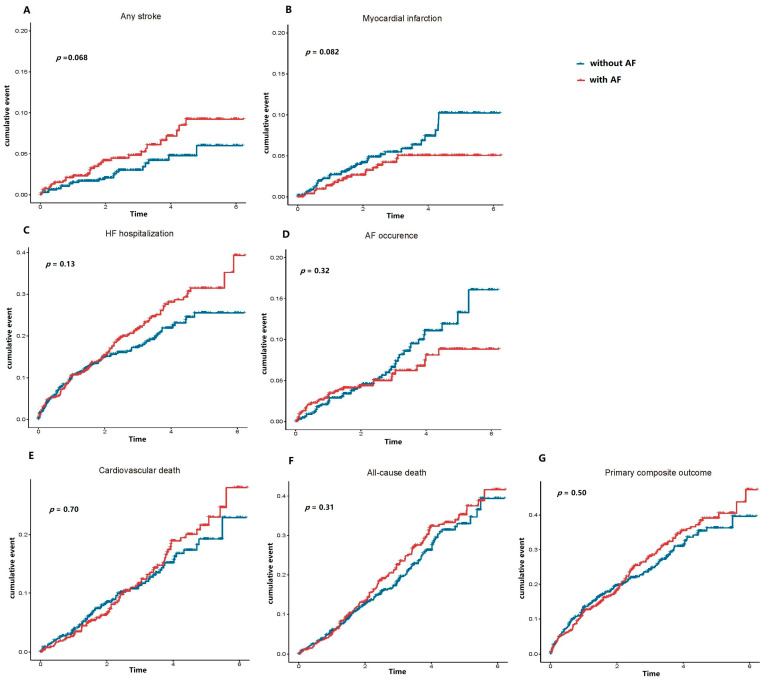
The cumulative probability of adverse outcomes prior to WRF between HFpEF patients with and without AF. Red line indicates HFpEF patients with AF, and blue line indicates HFpEF patients without AF.

**Figure 2 biomedicines-11-02484-f002:**
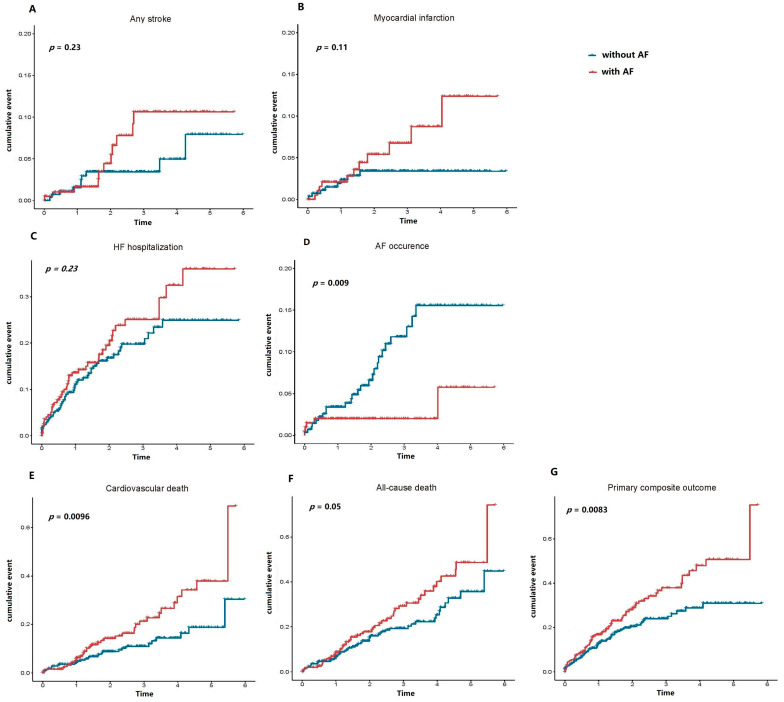
The cumulative probability of adverse outcomes post-WRF between HFpEF patients with and without AF. Red line indicates HFpEF patients with AF, and blue line indicates HFpEF patients without AF.

**Table 1 biomedicines-11-02484-t001:** Baseline patient characteristics of HFpEF patients with or without AF.

	Overall (n = 1763)	HFpEF without AF (n = 1004)	HFpEF with AF (n = 759)	*p*-Value
Demographics				
Age, yrs	71.5 ± 9.7	69.6 ± 10.0	74.0 ± 8.6	<0.001
Male, n (%)	884 (50.1%)	476 (47.3%)	408 (53.8%)	0.028
Race (white), n (%)	1382 (78.3%)	727 (72.3%)	655 (86.3%)	<0.001
Current smoker, n (%)	117 (6.6%)	86 (8.6%)	31 (4.1%)	<0.001
BMI, kg/m^2^	33.83 ± 8.2	34.4 ± 8.5	33.0 ± 7.6	<0.001
Physical examinations				
HR, bpm	69.1 ± 11.2	69.2 ± 11.6	68.9 ± 10.8	0.772
SBP, mmHg	127.5 ± 15.7	129.5 ± 16.1	124.8 ± 14.7	<0.001
DBP, mmHg	71.3 ± 11.5	71.7 ± 11.8	70.9 ± 11.0	0.095
EF, %	58.2 ± 7.8	58.6 ± 8.0	57.5 ± 7.4	0.002
eGFR, mL/min × 1.73 m^2^	60.7 ± 19.4	62.4 ± 20.5	58.6 ± 17.6	<0.001
Morbidities, n (%)				
WRF	756 (42.9%)	440 (43.8%)	316 (41.6%)	0.357
DM	788 (44.7%)	516 (50.8%)	278 (36.6%)	<0.001
Asthma	194 (11.0%)	122 (12.2%)	72 (9.5%)	0.158
HTN	1586 (90.0%)	913 (90.9%)	673 (88.7%)	0.149
Angina pectoris	485 (27.5%)	291 (29.0%)	194 (25.6%)	0.158
Previous MI	359 (20.4%)	221 (22.0%)	138 (18.2%)	0.08
Previous stroke	158 (9.0%)	84 (8.4%)	74 (9.7%)	0.312
NYHA functional class, n (%)				0.899
I–II	1141 (64.8%)	662 (65.0)	479 (64.6)	
III–IV	620 (35.2%)	354 (35.3)	266 (35.0)	
COPD	291 (16.5%)	163 (16.2)	128 (16.9)	0.697
ICD	42 (2.4%)	20 (2.0%)	22 (2.9%)	0.372
CABG	336 (19.1%)	200 (19.9%)	136 (17.9%)	0.222
PCI	344 (19.5%)	221 (22.0%)	123 (16.2%)	0.009
Pacemaker	242 (13.7%)	81 (8.1%)	161 (21.2%)	<0.001
Medications, n (%)				
Beta blocker	1386 (78.6%)	784 (78.1%)	602 (79.3%)	0.502
CCB	681 (38.6%)	390 (38.8%)	291 (38.3%)	0.94
Diuretics	1571 (89.1%)	874 (87.1%)	697 (91.8%)	0.002
ACEI/ARB	1393 (79.0%)	797 (79.4%)	596 (78.5%)	0.503
Aspirin	1026 (58.2%)	681 (67.8%)	345 (45.5%)	<0.001
Long-acting nitrate	305 (17.3%)	191 (19.0%)	114 (15.0%)	0.042
Statins	1148 (65.1%)	664 (66.1%)	484 (63.8%)	0.441
Warfarin	592 (33.6%)	59 (5.9%)	533 (70.2%)	<0.001

HFpEF, heart failure with preserved ejection fraction; TOPCAT, Treatment of Preserved Cardiac Function Heart Failure With an Aldosterone Antagonist; AF, atrial fibrillation; yrs, years; BMI, body mass index; HR, heart rate; SBP, systolic blood pressure; DBP, diastolic blood pressure; EF, ejection fraction; eGFR, estimated glomerular filtration rate; WRF, worsening renal function; DM, diabetes mellitus; HTN, hypertension; MI, myocardial infarction; NYHA, New York Heart Association; COPD, chronic obstructive pulmonary disease; ICD, implantable cardioverter defibrillator; CABG, coronary artery bypass grafting; PCI, percutaneous coronary intervention; CCB, calcium channel blocker; ACEI/ARB, angiotension converting enzyme inhibitors/angiotension Ⅱ receptor blocker.

**Table 2 biomedicines-11-02484-t002:** Incidence rates (per 100 patient-yrs) for adverse outcomes in HFpEF patients with and without AF.

Outcomes	Overall	HFpEF with AF	HFpEF without AF
Primary endpoint			
Prior to WRF	10.3 (8.6–20.0)	10.6 (8.0–13.2)	10.0 (7.8–12.2)
Post-WRF	12.1 (9.3–14.9)	16.5 (11.6–21.5)	9.3 (6.0–12.6)
All-cause death			
Prior to WRF	7.9 (6.4–9.4)	8.5 (6.1–10.8)	7.4 (5.5–9.4)
Post-WRF	9.4 (6.9–11.9)	11.5 (7.3–15.8)	8.0 (4.9–11.0)
Cardiovascular death			
Prior to WRF	4.4 (3.2–5.5)	4.6 (2.8–6.3)	4.2 (2.7–5.7)
Post-WRF	6.0 (4.0–8.0)	8.3 (4.6–12.0)	4.5 (2.2–6.9)
HF hospitalization			
Prior to WRF	7.4 (6.0–8.9)	8.2 (5.9–10.5)	6.8 (5.0–8.7)
Post-WRF	8.8 (6.3–11.2)	10.9 (6.7–15.0)	7.4 (4.5–10.4)
Stroke			
Prior to WRF	1.5 (0.8–2.1)	1.9 (0.7–3.0)	1.1 (0.4–1.9)
Post-WRF	2.0 (0.8–3.2)	2.6 (0.5–4.7)	1.5 (0.1–2.9)
MI			
Prior to WRF	1.7 (1.0–2.4)	1.3 (0.3–2.2)	2.1 (1.0–3.1)
Post-WRF	1.7 (0.6–2.9)	2.6 (0.4–4.7)	1.2 (0–3.1)
* AF occurrence			
Prior to WRF	2.3 (1.5–3.1)	2.0 (0.8–3.2)	2.5 (1.4–3.7)
Post-WRF	2.6 (1.2–4.0)	0.9 (0.3–2.2)	3.7 (1.6–5.9)

* AF = new-onset AF in HFpEF patients without history of AF or AF recurrence in HFpEF patients with AF; yrs, years; HFpEF, heart failure with preserved ejection fraction; AF, atrial fibrillation; WRF, worsening renal function; HF, heart failure; MI, myocardial infarction.

**Table 3 biomedicines-11-02484-t003:** Associations of worsening renal function with adverse outcomes in HFpEF patients with and without AF.

Outcomes	Overall		HFpEF with AF		HFpEF without AF		*p* for Interaction
	Unadjusted	Adjusted	Unadjusted	Adjusted	Unadjusted	Adjusted	
Primary composite outcome	1.30 (1.07–1.57)	1.29 (1.05–1.57)	1.64 (1.24–2.16)	1.58 (1.19–2.11)	1.08 (0.82–1.41)	1.10 (0.83–1.46)	0.008
All–cause death	1.28 (1.04–1.57)	1.36 (1.09–1.70)	1.36 (1.01–1.84)	1.50 (1.10–2.06)	1.22 (0.91–1.64)	1.24 (0.91–1.70)	0.450
Cardiovascular death	1.33 (1.01–1.76)	1.41 (1.06–1.88)	1.84 (1.26–2.71)	2.00 (1.34–3.00)	0.99 (0.66–1.48)	1.00 (0.65–1.53)	0.005
HF hospitalization	1.31 (1.05–1.64)	1.27 (1.00–1.60)	1.44 (1.04–1.98)	1.32 (0.94–1.85)	1.28 (0.94–1.73)	1.28 (0.93–1.76)	0.319
Stroke	1.05 (0.64–1.73)	1.05 (0.63–1.77)	0.82 (0.40–1.68)	0.82 (0.39–1.72)	1.45 (0.71–2.98)	1.49 (0.70–3.16)	0.361
MI	1.23 (0.79–1.90)	1.17 (0.75–1.85)	1.33 (0.66–2.66)	1.15 (0.55–2.41)	1.14 (0.65–1.99)	1.15 (0.64–2.06)	0.651
AF *	1.01 (0.68–1.50)	1.00 (0.66–1.50)	0.46 (0.17–1.21)	0.43 (0.16–1.16)	1.17 (0.74–1.83)	1.21 (0.76–1.94)	0.015

Adjusted for age, sex, race, treatment group, smoking status, NYHA functional class, BMI, SBP, diabetes mellitus, aspirin treatment, warfarin treatment, long-acting nitrate, diuretic therapy, history of PCI and pacemaker implantation. * AF = new-onset AF in HFpEF patients without history of AF or AF recurrence in HFpEF patients with AF; TOPCAT, Treatment of Preserved Cardiac Function Heart Failure with an Aldosterone Antagonist; HFpEF, heart failure with preserved ejection fraction; AF, atrial fibrillation; HR, hazard ratio; HF, heart failure; MI, myocardial infarction; PCI, percutaneous coronary intervention.

## Data Availability

Data are contained within the study.

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
