# Peer review of "Worsening Renal Function and Adverse Outcomes in Patients with HFpEF with or without Atrial Fibrillation"

_biomedicines, 2023, doi:10.3390/biomedicines11092484_

Round 1

Reviewer 1 Report

Overall a very good article that further analyzes the data from the well-designed TOPCAT study to demonstrate that worsening renal function, together with AF, is associated with a greater risk of adverse outcomes. The rationale behind the study, the population, methods and results are all clearly presented. An improvement to the discussion section could be made by adding a small paragraph regarding the practical implications of this study, especially taking into account the limitations mentioned. 

Additionally, please correct Table 3, which is too wide to fit and is cut-off in the final text. Also, the resolution of the figures could be improved.

Minor language corrections:

- Line 10: "are often coexist" -> remove "are"

- Line 129: "Af" -> "AF"

- Line 158: replace the full stop with comma, so that it becomes "When compared with non-AF patients post WRF, AF patients..."

- Line 228: "needs further studying

- Lines 236-238: needs better coherence, such as: "Finally, even though the participants in the TOPCAT study were evenly divided into the spironolactone or placebo group, we failed to further subdivide the two treatment groups due to limited data."

Author Response

Comments and Suggestions for Authors

Overall a very good article that further analyzes the data from the well-designed TOPCAT study to demonstrate that worsening renal function, together with AF, is associated with a greater risk of adverse outcomes. The rationale behind the study, the population, methods and results are all clearly presented. An improvement to the discussion section could be made by adding a small paragraph regarding the practical implications of this study, especially taking into account the limitations mentioned. 

Additionally, please correct Table 3, which is too wide to fit and is cut-off in the final text. Also, the resolution of the figures could be improved.

Response: Thank you for your comments. Now we have improved the Table 3 and the corresponding Figures.

Comments on the Quality of English Language

Response: Thank you for your suggestions. Now we have improved the language by using an English Editing service (https://languageediting.nature.com/en/). We also used the “grammarly” (https://app.grammarly.com/).

Minor language corrections:

- Line 10: "are often coexist" -> remove "are"

Response: Now we have revised it.

- Line 129: "Af" -> "AF"

Response: Now we have revised it.

- Line 158: replace the full stop with comma, so that it becomes "When compared with non-AF patients post WRF, AF patients..."

Response: Now we have revised it.

- Line 228: "needs further studying" 

Response: Now we have revised it. It should be “needs further investigation”.

- Lines 236-238: needs better coherence, such as: "Finally, even though the participants in the TOPCAT study were evenly divided into the spironolactone or placebo group, we failed to further subdivide the two treatment groups due to limited data."

Response: Now we have revised it according to your suggestion.

Reviewer 2 Report

First of all this is subanalysis of randomized trial, thus results of this study is related to significant bias

- How many patients were during dual antiplatelet theraphy? How many patients received NOACs?

Please add this study to disscusion an elaborate the topic

https://pubmed.ncbi.nlm.nih.gov/33708474/

- Rate of bleeding/thrombotic  complications?
- study is conducted without utilization of empagloflozin, which mighe increase kidney function and currently is recommended both in HFpEF and chronic kidney failure - nowadys more beneficial treatment is available, thus results of this study have low value

- Hyperuricemia is associated with the risk of developing atrial fibrillation and heart failure. Some studies suggested hyperuricemia as a marker of HFrEF, have authors observed similar results in HFpEF? Please add this study to discussion

https://pubmed.ncbi.nlm.nih.gov/35742536/

- what is clinical signifiacance of this study? What impact on everyday clinical practise?

thank you for opportunity to cooperate

Author Response

Reviewer 2:

Comments and Suggestions for Authors

First of all this is subanalysis of randomized trial, thus results of this study is related to significant bias

- How many patients were during dual antiplatelet theraphy? How many patients received NOACs?

Please add this study to disscusion an elaborate the topic

https://pubmed.ncbi.nlm.nih.gov/33708474/

Response: Thank you for your comments. In table 1, we have shown the data of aspirin and warfarin. We did not present NOACs because when the study (TOPCAT NCT00094302) began enrolling HFpEF patients (from 270 sites in 6 countries were enrolled between August 2006 and January 2012), the number of patients taking the drug was relatively small.

In the revised manuscript, we have cited the following study you provided:

Uzieblo-Zyczkowska B, Krzesinski P, Maciorowska M, et al. Antithrombotic therapy in patients with atrial fibrillation undergoing percutaneous coronary intervention, including compliance with current  guidelines-data from the POLish Atrial Fibrillation (POL-AF) Registry. Cardiovasc Diagn Ther 2021,11(1):14-27.

- Rate of bleeding/thrombotic complications?

Response: Thank you for your comments. We have presented the data of stroke in Table 2 and Table 3. However, no bleeding events and other thrombotic complications were provided in the TOPCAT trial.

- study is conducted without utilization of empagloflozin, which mighe increase kidney function and currently is recommended both in HFpEF and chronic kidney failure - nowadys more beneficial treatment is available, thus results of this study have low value

Response: Thank you for your comments. Empagloflozin has been recommended in patients with HFpEF. However, the TOPCAT study ( NCT00094302) did not enroll patients with empagloflozin. As such, we can not provide the data analysis of empagloflozin.

- Hyperuricemia is associated with the risk of developing atrial fibrillation and heart failure. Some studies suggested hyperuricemia as a marker of HFrEF, have authors observed similar results in HFpEF? Please add this study to discussion

https://pubmed.ncbi.nlm.nih.gov/35742536/

Response: Thank you for your comments. The TOPCAT study ( NCT00094302) did not the baseline data of uric acid. As such, we can not provide the data analysis of hyperuricemia on our results, which needs further examination.

In the revised manuscript, we have cited the following study you provided:

Welnicki M, Gorczyca-Glowacka I, Lubas A, et al. Association of Hyperuricemia with Impaired Left Ventricular Systolic Function in  Patients with Atrial Fibrillation and Preserved Kidney Function: Analysis of the POL-AF Registry Cohort. Int J Environ Res Public Health 2022,19(12).

- what is clinical signifiacance of this study? What impact on everyday clinical practise?

Response: Thank you for your comments. As a common comorbidity of HFpEF, AF has a strong correlation with age and shares common clinical features with HFpEF. In addition, the presence of worsening renal function (WRF) is common in HFpEF patients. In our current study, WRF was correlated with a poor prognosis in HFpEF patients, especially for those with AF. Thus, our results indicated that HFpEF patients with AF need to pay attention to the management of renal function in everyday clinical practice.

Reviewer 3 Report

Line 32: heart failure is not necessarily terminal, especially when symptoms are of NYHA class II.

Line 132: were there no patients taking direct oral anticoagulating medication (which is superior to warfarin)

Line 147: this strongly suggests that patients with paroxysmal AF were involved. Where there also patients with permanent AF and patients treated with ablation? This statement is also very counterintuitive. With the statements of lines 215-217, one can expect AF to be an progressive conditions, where recurrences become more likely after prior events.

What is known about diastolic LV dysfunction (for example e/é) and LA size in these patients? Data about LV hypertrophy and its degree are absent

Is anything known about BNP levels during and after WRF?

The use of English language is acceptable.

Author Response

Reviewer 3:

Comments and Suggestions for Authors

Line 32: heart failure is not necessarily terminal, especially when symptoms are of NYHA class II.

Response: Thank you for your comments. Now we have deleted the word of “terminal”.

Line 132: were there no patients taking direct oral anticoagulating medication (which is superior to warfarin)

Response: Thank you for your comments. We did not present NOACs (non-vitamin K antagonist oral anticoagulants) because when the study (TOPCAT NCT00094302) began enrolling HFpEF patients (from 270 sites in 6 countries were enrolled between August 2006 and January 2012), the number of patients taking the drug was relatively small. We have stated it in the limitation section.

Line 147: this strongly suggests that patients with paroxysmal AF were involved. Where there also patients with permanent AF and patients treated with ablation? This statement is also very counterintuitive. With the statements of lines 215-217, one can expect AF to be an progressive conditions, where recurrences become more likely after prior events.

Response: Sorry for the confusion. We actually found this phenomenon: the cumulative probability of AF prior to WRF was comparable between HFpEF patients with and without AF, whereas the cumulative probability of AF post WRF was higher in HFpEF patients without AF compared with those with AF. However, these findings did not remain after adjustment (Table 3). We have revised the corresponding description in the revised manuscript.

What is known about diastolic LV dysfunction (for example e/é) and LA size in these patients? Data about LV hypertrophy and its degree are absent

Response: Thank you for your comments. The TOPCAT trial included 3445 symptomatic HFpEF from 270 sites in 6 countries were enrolled between August 2006 and January 2012. However, echocardiograms were obtained from 935 patients with HFpEF. In the previous study by Maja Cikes et al, the authors have presented the echocardiographic data between HFpEF patients with and without AF using the TOPCAT data (following Table 2). As a result, ultrasound data are not reproduced in our current study.

Cikes M, Claggett B, Shah A M, et al. Atrial Fibrillation in Heart Failure With Preserved Ejection Fraction: The TOPCAT Trial[J]. JACC Heart Fail, 2018,6(8):689-697.

Is anything known about BNP levels during and after WRF?

Response: Thank you for your comments. Unfortunately, in the TOPCAT trial, we could provide the baseline BNP levels, but BNP levels during and after WRF were not shown in this trial.

Comments on the Quality of English Language

The use of English language is acceptable.

Response: Thank you for your suggestions. Now we have improved the language by using an English Editing service (https://languageediting.nature.com/en/). We also used the “grammarly” (https://app.grammarly.com/).

Round 2

Reviewer 2 Report

no further comments

-

Author Response

Thank you for your review

Reviewer 3 Report

 The limitations should state also the following

1) Ultrasound studies are not reproduced: e/é and other parameters for diastolic function are absent

2) The same applies to the BNP during and after the worsening of renal function

Author Response

Thank you for your comments. Now we have added these contents to the limitations.